# Australian Hospital Experiences of People Living with Deafblindness or Dual Sensory Impairment: The Report Card

**DOI:** 10.3390/healthcare12080852

**Published:** 2024-04-17

**Authors:** Annmaree Watharow

**Affiliations:** Centre for Disability Research and Policy, University of Sydney, Sydney, NSW 2006, Australia; annmaree.watharow@sydney.edu.au

**Keywords:** quantitative, deafblind, dual sensory impairment, hospital experiences

## Abstract

Gaps abound in the literature about what happens when people living with deafblindness or dual sensory impairment (DBDSI) go to the hospital. Anecdotally, from my lived experiences and professional work, as well as from within communities, stories are told about how hazardous it is to be a patient in an Australian hospital for those living with DBDSI. This paper outlines a quantitative component of a mixed-methods study examining the intricacies of these experiences. The research objective was to discover what hospital interactions looked like for patients living with DBDSI. A constrained question set was used, namely, the Australian hospital experience question set (AHPEQS 2017). It asked patients about key factors in their hospital interactions. The results form a distressing snapshot of care and communication interactions. Experiences of flouting protective conventions, dehumanisation, neglect, discrimination, disparate care, inaccessible consent forms, and a lack of communication predominate. The participants reported experiences from multiple different hospitals, so these findings suggest a broad culture of failing to provide patient-centred care and accessible-to-the-patient communication. The findings showcase the urgency for more research and remedial actions to be undertaken by both professionals and institutions.

## 1. Introduction

Little is known about the hospital experiences of those living with deafblindness, or dual sensory impairment, from their standpoint, in their own ways of telling. I found nothing in the academic literature in the Australian context. Yet, sensory loss is remarkably common in the community, both nationally and globally. The likelihood of experiencing sensory loss increases with age, and it is often an “invisible” disability. The three most common types of sensory loss are vision loss, hearing loss, and a combined sensory loss, most often called deafblindness or dual sensory impairment (DBDSI).

Contestations and debates about terminology plague researchers and practitioners in this field, with a fairly even split between the two terms [1]. This occurs with vacillation between identity-first language—a deafblind person, for example—and person-first language—a person with deafblindness or a person living with dual sensory impairment, for instance. Other terms in use include dual sensory loss and multi-sensory loss/impairment, while some may identify as a person with blindness and hearing loss or a deaf person with low vision. There are many permutations of language in use to describe impairments and losses. Recent work suggests that, notwithstanding the person-first usage in protections and legislation (for example, the Americans with Disability Act 1992, the United Nations Charter on the Rights of People with Disability (UNCRPD) 2006, and the National Disability Insurance Scheme Act (Australia) 2013), there is generally a mélange of both identity-first and person-first language in common usage among individuals with disability [2,3,4]. (Incidentally, a United States study across three countries found that 42% of people living with disability preferred identity-first language, 38% preferred person-first language, and 18% were comfortable with either nomenclature [5].

### 1.1. Position Statement

I suggest an inclusive term, “deafblindess–dual sensory impairment”, with the acronym DBDSI. In this paper, person-first language is used, that is, a person with deafblindness–dual sensory impairment, for example. These terms are used to refer to people with co-occurring sight and hearing loss. I accept that there are many identifiers that individuals use, whether person-first, identity-first, or other nomenclature, and strongly advise that researchers and writers provide a position statement (as above) that clearly illustrates any umbrella terms and identifiers. I occupy the position of a lived-experience researcher and am the lead of this project. (The occasional use of an inclusive “we” reflects the participants’ position as expert knowers and research partners.) I live with Usher syndrome, that is, congenital deafness with blindness/low vision due to retinitis pigmentosa, which usually occurs in the first or second decade of life. It is a degenerative condition that may also be accompanied by a balance disturbance.

This paper will outline, in brief, definitions and data around DBDSI, followed by exploration of the general absence of hospital experience data for this population in Australia, and then finally present the research that is undertaken to fill the gap. Particular attention is paid to the methodological challenges of this project to examine the hospital experiences of people with DBDSI. The commitment to inclusion and accessibility for all in the research space is observed and outlined. I then present the results and discuss them and their implications.

### 1.2. About DBDSI

As a concept, DBDSI is less easy to define than single sensory loss, and there is vast heterogeneity in impairments and functions. Co-occurring hearing and vision loss are disproportionately cumulative in their effects. Yet, despite the detrimental impact of DBDSI on communication, access to information, and mobility/orientation, it remains an invisible, poorly supported, and complex disability. DBDSI has been called the “third sensory loss“ to highlight its significance and distinct status [6].

The Nordic definition of DBDSI proves useful (it should be noted that deafblindness is used here as an umbrella term for the co-occurrence of sight and hearing loss):

Deafblindness is a distinct disability. Deafblindness is a combined vision and hearing disability. It limits the activities of a person and restricts their full participation in society to such a degree that society is required to facilitate specific services, environmental alterations, and/or technology [7].

This definition comes with a comprehensive set of explanatory notes that outline many of the key complexities of deafblindness. These are summarised below [8]: It is a distinct and complex disability;It is hard for one sense to compensate for the loss of the other;It is time-consuming;It is energy-draining;It is activity-limiting;It is participation-reducing;Information is received in fragments;Communication, access to information, and mobility are adversely affected;Tactile sense is critical as a conduit of information;Communication technology, assistive devices, interpreters, and adaptations to the environment may be required;Human assistance and support are often critical to access to information, mobility, and safety;Society is obliged to fund/provide all necessary support. These complexities are important because they impinge on every aspect of life, including healthcare and hospitalisation.

## 2. Prevalence

Single sensory loss is very common, increasing with age. Globally, the World Health Organization (WHO) estimated that there were 1.5 billion people (or about 20% of the global population) living with impaired hearing in 2021. The estimated prevalence of hearing loss is expected to increase to one in four people by 2050 [9]. In Australia, hearing loss is the third largest cause of years lived with disability, affecting more than 60% of people over 60 years of age [10]. In terms of vision loss, the WHO estimated that 2.2 billion people in 2021 had impaired vision due to disease or uncorrected refractive errors [11]. Vision impairment also increases with age. In an Australian context, vision loss impacts 5% of non-indigenous Australians aged 50–59 years old but increases to over one-third by age 90 and over [12]. Separately, each sensory loss can compromise interaction with the environment and access to information and contribute to communication disruption.

DBDSI is highly prevalent in older adults. Data, if available at all, amount to estimates. Using a multi-country survey, the World Federation of the Deafblind estimates that roughly between 0.2 and 2% of the global population live with DBDSI [13]. Few countries, including Australia, collect data specifically on DBDSI as a distinct disability, which obscures true statistical visibility. Thus, accurate measurement and reporting on DBDSI are not available. Therefore, the current estimate that one in four older Australians over the age of 80 live with DBDSI is likely a significant underrepresentation [14].

As stated earlier, the combination of both hearing and vision loss limits the capacity of one sense to compensate for the other, with devastating and far-reaching impacts in all areas of life—particularly in terms of access to information, communication, safe mobilising and orientation, sense of disconnection and social isolation, and vulnerability in some situations, such as hospitals. The majority of people with DBDSI are older citizens, and they are often poorly served by social support and aged-care institutions [15]. Social exclusion and isolation are common to the DBDSI experience, which has a human cost for the individual and their families as well as social implications for their health and well-being, likelihood of misadventure, and carer challenges [16]. Much of this human suffering is veiled, with limited targeted research, service provision, or support directed to it. Older adults with DBDSI are a marginalised population group. 

Existing literature shows sensory loss is associated with more healthcare encounters, higher rates of hospitalisation, poorer outcomes, and higher rates of communication breakdown. All these cumulatively create a significant cost to the community [17,18,19]. Patients living with sensory loss report greater dissatisfaction with many aspects of their care, especially with the information provided by caregivers [20]. Even though largely absent from the research literature, discussions around negative hospital and healthcare experiences are common in community and disabled people’s organisations.

### 2.1. The Literature

Currently, DBDSI, particularly in older people where the condition is most prevalent, is under-recognised in Australia [21]. There is a dearth of literature on DBDSI generally, but there is a growing research and practitioner community. Literature and academic research on the lived hospital experience of people with DBDSI is even more sparse, with a conference paper from Takahashi (2019) of Japan noting restrictive practices, a lack of awareness, and few deafblind specialist services being common to the experiences of DBDSI patients and their families [22]. A Spanish report found lack of access, orientation, and mobility generally for outpatient hospital encounters but did not explore inpatient experiences [23]. Lived-experience accounts of mostly negative healthcare encounters from eight contributors with a DBDSI called “Bad Medicine” (including hospital stays) are found in Stoffel’s Deafblind Reality [24]. Mascia and Silver (1996) describe a case study of a person with deafblindness who had an anaesthetist who fingerspelled the manual alphabet, which made for a positive hospital stay [25]. Sense UK’s 2016 report, Equal Access to Healthcare, notes repeated issues with inaccessible formats, a lack of information, and ill-informed professionals. Health systems were found to be not complying with the United Kingdom’s accessibility legislation [26]. No hospital-specific material was found from the Australian context, although healthcare experiences are mentioned as concerns in broader research on the social aspects of living with dual sensory impairment and life course stories [16,27]. Gaps abound too in our knowledge from the point of view of people with DBDSI, that is, knowledge constructed with them or by them, not observations made about them by outside observers such as professionals. 

### 2.2. Research Question 

This lived-experience-led project asks what the hospital experiences of people with DBDSI are and endeavours to come closer to the patient from a DBDSI standpoint through coproduction, cultural immersion, and the provision of individual accessibility. The project to find answers is a mixed-methods study: (1) a quantitative component using an Australian patient experience question set and (2) a qualitative inquiry of semi-structured interviews asking, “What happened when you went to hospital?” This paper focuses on the quantitative component exclusively. 

### 2.3. Theoretical Framing 

I use an epistemic justice framing [28]. Epistemic injustice occurs when participants are wronged in their capacities as expert knowers, that is, their experiences are not sought, made legible, counted, or acknowledged [28]. It must be acknowledged that there cannot be true knowledge building if we are not approaching our research in socially just ways. This means actively seeking out the marginalised and providing full accessibility and accommodations. 

There are two key injustices in knowledge building: Testimonial injustice;Hermeneutic injustice.

People with DBDSI experience both of these alone and in combination. They may not be asked for their expert knowledge, as researchers may not provide accessible formats or interpreters. Biases may lead to devaluation and stereotyping. Ageism and ableism exert power over whose stories are sought, heard, understood, and disseminated. Older people may believe that their co-occurring sensory loss is a natural part of ageing, so they are hermeneutically unable to name their experiences or to help themselves improve their situation. Accessibility is critical to uncovering experiences and promoting social and epistemic justice. Fricker notes the failure to provide accessibility points “to the distinctive intelligibility disadvantage experienced by those who speak, write, and sign differently from what is expected/habitual in the context provided” (personal communication, Fricker, 2023). This present paper demonstrates data collection and knowledge building that actively seeks and supports participants with DBDSI so that their experiences are rendered legible. 

## 3. Methodology

This paper examines the quantitative component of a mixed-methods study in which a patient experience survey is adjusted so that participants with DBDSI can access it in the communication mode that works for them. 

Surveys are perhaps the most common research tool found across the medical, health, social sciences, and humanities disciplines. They are the mainstay of an increasingly digital world. Digital ways of researching provide opportunities for reaching many people with disability but largely exclude those with DBDSI. Surveys, whether online or on paper, are simply not accessible for all with DBDSI, whether as participants or as lived-experience researchers, such as me. Understanding the accessibility needs of this group is vital to enabling voices and signs to be heard, seen, and felt, and then counted to show what is happening in hospitals. Without providing full accessibility support, we would be guilty of excluding people with DBDSI from research, thus contributing to erasure and epistemic injustice [28]. For people with DBDSI, this means researchers must navigate accessibility barriers and provide safe spaces for conducting research. Careful attention to the complexities of communication is required, as is providing alternative ways of accessing information and ensuring the safety and orientation needed when navigating the research environment and any travel involved. The use of lived-experience researchers in project teams is an effective, emancipatory way to bridge this knowledge gap. By bringing lived experiences directly to research leadership, researchers and team members can move a project closer to the insider position and reduce the likelihood of epistemic injustices [29].

To address the question of what the Australian hospital experiences of people are for people with DBDSI, 18 self-identified people with DBDSI were asked to respond to a constrained set of questions from an adapted standard questionnaire from the Australian Hospital Patient Experience Question Set (AHPEQS) [30]. A semi-structured narrative inquiry interview was also conducted with each participant, which will be discussed in a separate future article.

The AHPEQS includes 101 factors of hospital care, distilled into 12 questions, and was designed with Australian consumers in mind. The 2017 version was used in this study as the AHPEQS had no alternative formats until 2020, when a large print version was added. Again, it is still not accessible to all. Each participant in this project was therefore asked what format(s) would best suit them, and the researcher provided these. This meant giving participants choice and control over the location; the communication mode(s) used; and the presence or absence of support workers, carers, or partners. They were also able to select an interpreter of their choosing, the technological aids they wanted, and alternative formats for information sheets and consent forms.

There are other methodological and accessibility challenges to using the AHPEQS [30]: Hierarchical scales are an accessibility barrier for this population. People with DBDSI find these confusing [31], and not all accessibility software translates them accurately. It is challenging for tactile interpreters to unpack all the components of such scales [31,32]. Therefore, we eliminated these from the research question set. This means that the results indicate if a factor was present or not but could not determine its magnitude.In Roy’s (2 work on better practice research for people living with DBDSI, the difficulties of unpacking two-part questions for interpreters and participants are noted, especially when these are unrelated concepts [31]. An example of this in the AHPEQS is a question that reads, “Is the hospital clean and welcoming?” We separated these into two individual questions:(A)Is your hospital welcoming?(B)Is your hospital clean?All the participants had low vision/blindness and commented that they were unable to say if a hospital was clean or not. Given this, we eliminated this question from the evaluation.We further adapted the AHPEQS for this research to include a specific question about accessible consent forms: “Did you receive an accessible-to-you consent form at the hospital?” This was, to our mind, a glaring omission from the original AHPEQS questionnaire.

### 3.1. Conduct of the Study

The study was helmed by a lived-experience researcher and clinician. I believe that researchers must collaborate in co-production with people with DBDSI (and indeed all disabilities) in the construction of knowledge about their experiences, lives, and emotions. I also believe that research must benefit the community from which it arises. There was co-production and cultural immersion with members of sensory impairment support groups for 18 months before the onset of data collection. This gave crucial insights into how communication might work and what kinds of adjustments and accommodations would be needed for such a heterogeneous group. Stone and Priestley (1996) note that emancipatory research requires researchers to adopt a multitude of methods for data collection so that the diverse needs of participants with disability (and lived-experience researchers) can be accommodated [33]. Failing to do so is to continue to exclude and erase.

Participants were recruited from three disabled people’s organisations and also by “snowballing” [34]. Silicone wristbands were provided as an incentive. These stated various identities, such as “I am deafblind”, “I need an interpreter”, “I have low vision”, “I have hearing loss”, and “I have dual sensory impairment”, among others. Some participants chose two bands, e.g., the combination of “I have hearing loss” and “I have low vision”. These reflect the complexities of identity in the heterogeneous group.

The inclusion criteria were limited to ensure no one who wished to participate was excluded:Self-identified as having co-occurring hearing loss and low vision of any type;Adults over 18 years.

Each participant taking part in the study had fully accessible-to-them support at each stage of the research process. Most participants required multiple communication methods. Methods included combinations of speech, text, email, large print, extra-large print, braille, sign language (restricted frame), tactile sign language, deafblind manual alphabet and fingerspelling, idiosyncratic tactile languages, interpreters, and various assistive software and devices. For data collection, most preferred face-to-face methods. 

The lived-experience researcher also had fully accessible-to-them support at each stage of the research process funded by the University of Technology Sydney. The researcher was supported by an accessibility assistant whose function was to supplement communication with tactile signs and messaging, take their own notes, and liaise with all parties in setting up meeting places and times among the people involved. The accessibility assistants were also responsible for environmental adjustments to rooms depending on participant preferences, for example, dimming the room’s lighting for those with sensitive retinas or contrast/glare sensitivity or increasing room light for those with residual vision when requested. Seating also had to be arranged with tactile tables as needed, interpreters and participants sitting in comfortable proximity for tactile signing, and safe places for service animals to snooze during question time.

This meant, in practice, that the research space expanded in physical ways. For example, a larger room was needed for face-to-face data collection, and there were more bodies and voices in the room. Therefore, ensuring each participant was the principal voice became central to rendering these experiences visible. With the exception of data collection by email/online by one participant, all others required extra bodies, for example, national relay service operators, interpreters, support workers, communication guides, partners, and/or carers. In three instances, service animals were also present.

One-third required interpreters to collect responses to the AHPEQS. These participants gave the researcher a list of their three most trusted interpreters. I intended and succeeded in engaging the first preference for each. Ahead of the data collection activities, we met with interpreters to go through the research question set; this was how the difficulties mentioned previously were revealed. This meant we could mostly adjust the research question set for clarity and access ahead of time.

All of this did, however, mean that another research challenge, a temporal challenge, emerged. The research began in 2018 and was completed in 2021. It took far longer to organise events than for sighted-hearing research participants, who could use print-based/online surveys unassisted, and to find mutually convenient meeting times—for not just the researcher and participant but also for the additional support personnel needed to confer a sense of safety and trust, along with full individual accessibility. Extra time was needed for the unpacking and clarifying of questions in situ, relaying (which is also noted by Hersh, 2013a), and rest breaks [35]. Tactile languages are kinetic, and interpreters have mandated rest breaks that participants also benefit from [36]. While this may sound like the question flow was interrupted, this is a normal social experience for people who use interpreters for visual and tactile sign communications, as was most evident during immersion in support groups, where two interpreters would switch every ten minutes. If there was only one interpreter per attendee, the whole room would wait before restarting after mandated rest breaks.

We should also mention here the unavoidable fact that providing accessibility as well as safe, supported research spaces for all requires more funding than not providing accessibility and safety. It is our view that academic institutions, ethics departments, and grant/funding bodies need to require accessibility and safety plans for all research endeavours and budget for these accordingly. For this research, however, the researcher self-funded the accessibility requirements for the participants, while the university provided all the researcher’s support needs during the research process. Information and informed consent materials were provided in multiple formats, including plain language, braille, large print, very large print, electronic, and hard copies. Video Auslan materials were available but not requested by this group, and very large print was the most common format requested. Ethics approval was obtained (No. HREC ETH17-1398).

### 3.2. The Participants

Below, in list format, are some demographic particulars and impairment types of the participants:Age range: 25 to 71 years;Under 65 years: 14 participants;Over 65 years: 4 participants;Male: 5 participants;Female: 13 participants;Other: none identified;None identified as Aboriginal or Torres Strait Islander people;Resident in an aged care facility: 5 (although 2 of these were under 65 years old and represent a failure by society to provide safe housing for younger people with DBDSI);Urban: 14;Rural: 4;Remote: 0.

#### Identity

This is complex territory. Most participants had multiple identities depending on the places and people they were with, for example, “I say deafblind in hospital but when I’m at work I say I am Deaf with a bit of sight loss.” The capital D here denotes that the participant identifies as culturally Deaf. Deafblind identity and culture were strongest amongst the sign language users: “Deafblind and bloody proud of it”, said one. Some participants felt hospital staff paid more attention to them if they said they were deafblind rather than dual sensory impaired. We asked what identity they used in hospitals and healthcare settings:Deafblind: 12;Dual sensory impairment: 4;Other: 2 (I have low vision and I am deaf/I am blind but I have hearing loss too).

Interestingly, given other work on identity language (see [5,37]) this study had very low rates of person-first identity. None said, “I live with (impairment)”. Most participants said, “I am (impairment)” or “I am (impairment and impairment)”. None said, “I am a person with (impairment)”.

### 3.3. Multiple Disability

Although participants were not asked about multiple disabilities in the demographic questioning, 8 participants volunteered this information.

### 3.4. Cultural and Linguistic Diversity 

Tactile/hand-over-hand/restricted frame/visual sign language (Auslan) users: 8;Nonverbal: 4.

#### 3.4.1. Impairment Information

Congenital deafblindness: 3;Usher syndrome (congenital hearing loss with acquired low vision): 9;Congenital hearing loss with acquired low vision: 1;Congenital blindness with acquired hearing loss: 1;Acquired hearing loss and low vision: 4.

This is a heterogenous group, but the group does not reflect the predominance in the general population of the over-65-year-old group with acquired causation. However, as Fricker (2011) has noted in his work on epistemic injustice, it is the people most impacted by failures of social structures that will have the most to say about these [28]. These participants, the majority of whom have lived a very long time with DBDSI, will accordingly have the most experience to offer to inform us of hospital and professional performance, provided they are given full accessibility and communication support. Otherwise, these experiences remain silenced—another epistemic injustice.

It is also important when discussing impairment and disability for us to understand that these impairment types and conditions are not necessarily singular or collective [38]. For example, two of the people with Usher syndrome also had other conditions affecting their sight or hearing. Other people have more than one eye condition, for example, macular degeneration plus other ocular vascular diseases. Cataracts were a very common comorbidity, as was presbycusis.

#### 3.4.2. Hospitals

Participants spoke about their most recent hospital stay in the quantitative component. All were admitted for stays longer than 24 h. 

Sixteen were public hospitals (Australia has universal healthcare in a two-tier system: public and private);Two were private hospitals;Fourteen hospitals were in urban settings across three states;Four were in regional centres.

#### 3.4.3. Analysis

Answers were collected from participants in their chosen communication mode and collated as factors present or not present. 

## 4. Results

Factors affecting the quality of participant experiences in hospitals were from the AHPEQS [30]. The categories in the question set were as follows:Interpersonal interactions;Clinical quality interactions;Care delivery interactions;Administrative interactions;The additional consent interaction question.

The results from each category are listed below. The quantitative results are presented in list form and in written description and are specifically not presented in table formats to enable better access by those with limited vision and those who use screen readers to access information.

Interpersonal interactions:
I am heard = 6%;I am cared about = 11%;I am known = 11%;I am treated as a human being = 17%;I understand what professionals say = 11%;I am informed = 44%.Clinical quality interactions:
I can get the right care at the right time = 17%;I experience high-quality and safe clinical care = 22%.Care delivery interactions:
I have confidence in the professionals treating me = 22%;I am discharged at the right time with the right plan = 50%;My personal care needs are attended to = 11%;My care is tailored to my needs = 22%;Different parts of my care are coordinated = 17%;I am treated equally no matter who I am = 22%.Administrative interactions:
My hospital puts the needs of patients first = 6%;My hospital is well managed overall = 6%;My appointments and waits are well managed = 0%;My feedback is welcomed and acted upon = 6%;My health records are well managed = 11%;My hospital is welcoming = 6%.Consent interactions
I receive an accessible-to-me consent form = 0%.

## 5. Discussion

The quantitative component of a mixed-methods project outlined here illuminates the breadth of the overwhelmingly negative experiences of people with DBDSI in Australian hospitals. For the depth, specificities, and complexities, it will be necessary to examine the qualitative data in a later article. What is evident from the above report card is that Australian hospitals broadly fail to meet the care and communication needs of participants with DBDSI.

Living with DBDSI is complex, and as noted elsewhere in this paper, more than the sum of the sensory loss parts. One component of these complexities is that people with DBDSI are at greater risk of needing hospital care and attention as a result of a range of social, health, and situational risk factors as well as the hazards posed by the impairments themselves [8]. Multiple disabilities are a common living reality for many people with DBDSI and range from 20 to 75% depending on age [13,39]. The present participant group is no exception, with a high rate of volunteering multiple disability information (44%). All this means that staff in hospitals must be patient-centred and attuned to the individual care and communication needs of patients with DBDSI (and indeed, all patients). From the results, this is clearly not happening, and the failures are present across all hospitals and types. 

The report card demonstrates there is a flouting of protections; a failure to provide communication, information, and informed consent; and unacceptable rates of dehumanisation and neglect. No hospital interaction scored a “pass“ grade. We look now at the conclusions derived from the APHPEQS data. 

## 6. Flouting of Protective Conventions

As mentioned above, the first significant conclusion we can draw from the report card is the flouting of protections. Numerous protective conventions exist and are designed to safeguard people who live with disability both generally and specifically in hospital settings (UNCRPD; Australian federal, state, and territory anti-discrimination and disability support legislation, e.g., Disability Services Act (Cth) 1986; etc.). These clearly enshrine the rights to safe care, equitable health services, information, and quality communication. The report card shows these are missing in action; the absence of care and communication is not remarkable or audited, nor do consequences arise for professionals or systems from this failure.

There are, clearly, major power disparities if protections and policies designed to keep patients safe in hospitals are ignored. It is necessary, then, to establish how and why power is exercised by hospitals/institutions and the professionals that work there. Hospitals, like other institutions, reflect settings where there is an uneven power gradient between patients and caregivers. One element of this is knowledge, where hospitals/institutions and their professionals control access to knowledge—of the environment, health conditions, and medical treatments. The withholding of knowledge is an exercise in power [40]. The situation is exacerbated for people with DBDSI because of the impact sensory loss has on access to knowledge. Patients are mostly in the position of not knowing what is going on because they are reliably uninformed by healthcare professionals and unreliably informed by personal support networks. This hazardous combination undermines ontological security for the individual and generates anxiety.

This anxiety is in addition to the accompanying stress of disease or injury that precipitates hospitalisation [41]. Anxiety, a lack of trust, and distress can lead to fear. Shared decision-making is diminished in the presence of fear and when there is a lack or restriction of access to information. These will need to be unpacked in the narratives of a qualitative inquiry. 

What this report card shows us is how ingrained and pervasive the flouting of those protections is. There are failures in every single interaction type of the AHPEQS. 

We will now examine the five interactive categories individually. 

### 6.1. Interpersonal Interactions

This category is the most fraught, offering a compelling insight into dissonance, communication failures, and dehumanisation.

#### 6.1.1. Dissonance 

There is an apparent dissonance in the interpersonal interaction space. Most participants (89%) felt they did not understand what professionals said; yet, 44% agreed that they were informed. One of the benefits of conducting surveys like the AHPEQS in personally negotiated, accessible ways is that we can go back and query such dissonance. Upon so doing, we discovered that participants had understood this question to mean that somebody, anybody, but not the professionals, had conveyed enough information to them for them to feel informed. The participants listed partners, carers, support workers, the Internet, and/or interpreters as non-hospital bodies who were keeping them informed. Over half of the participants felt uninformed by everybody. These results strongly suggest that, for participants with a communication disability, professionals are failing to communicate with most of them. Further, the AHPEQS question of “being informed” is ambiguous by not clarifying by whom they are informed.

#### 6.1.2. Communication Failures

A failure to communicate and inform is evident, given 89% of participants did not understand the healthcare professionals treating them, and 56% were not informed by anybody else, such as their support network or the Internet. Effective communication is the lynchpin of patient-centred care doctrines and shared decision-making models. The obvious inference from the very low rates of participants understanding their healthcare providers means that accommodations for their communication needs were either denied or ignored. In short, they were overlooked.

Good communication is reflected in asking patients what they need and providing them with accessible information, which is critical to better health and well-being. Good communication confers benefits; it allows patients to know what is going on, promotes a feeling of security and safety, and contributes to better health outcomes. It is an expression of the promise of equitable care that legal protections, policies, and communities expect.

Good communication also confers benefits beyond the state of “knowing what is going on”, including the following [42]: Shorter length of stay;Fewer hospital readmissions;Reduced emergency room visits;Closer treatment adherence;More effective medical follow-up;Less unnecessary diagnostic testing;Improved healthcare outcomes.

Communication and “knowing what is going on” are essential to building trust, security, and well-being [43,44,45]. Ontological security is the sense of trust and predictability in people, places, and things, which is necessary to feel safe and have a sense of coherence [44,46]. The sensory loss of DBDSI already exerts an unsteadying influence on the sense of safety and coherence, meaning that predictable and communicative people are even more critical to the maintenance of ontological security for people with DBDSI [43]. This, in turn, means that hospitals and professionals need to do more to ensure they are promoting communication and safety. On the other hand, poor communication between professionals and patients in hospitals can confer risk, and communication failures underpin nearly all adverse events in Australian hospitals [42]. 

There is a propensity for healthcare providers and institutions to consider communication failures to be solely a consequence of sensory loss. For example, “the patient didn’t understand their discharge medications because they are deaf.” Rather, communication failures are the result of the healthcare professional’s own attitudes, a poor disability knowledge base, and a lack of accessible information forms and supports [35,47]. The responsibility for communication in hospitals belongs to the healthcare professionals who hold knowledge—of the environment, health conditions, and medical treatments. This means when a patient with DBDSI (or indeed any communication disability) is admitted, they need to be identified, then asked if they need to make communications work. Communication assistance is then provided. This critical information should be handed over to all involved in the patient’s care so that the poor performance of healthcare professionals and negative experiences of patients with DBDSI are avoided.

#### 6.1.3. Dehumanisation

The results are not just indicative of communication and legal protection failures but also reflect human rights abuses. Dehumanisation is terrifyingly evident, with 83% of participants stating they were treated as less than human beings. This points squarely to carers being guilty of attitudes and practices that dehumanise, whether intentionally or otherwise, and shows that the existing protections and policies to prevent these are not being enforced.

## 7. Clinical Quality Interactions

It is not surprising, given the results in the other categories of the report card, that clinical quality interactions are poor. Most participants clearly lack confidence in the clinical care, with 78% saying they did not feel they received quality and safe care. Work on ontological security as mentioned above (i.e., confidence, trust, and predictability in people, places, and things) is paramount to being able to solve problems and navigate the hazards of life [46]. This is especially true for people with DBDSI who need trusted humans around them to confer confidence and safety [43]. For any patient, being in the hospital is an ontologically challenging experience. For sighted-hearing individuals, though, there is a much greater chance of seeing and hearing information and being able to query their care, which confers confidence. 

## 8. Care Delivery Interactions

Disparities in care delivery are alarmingly present, with neglect and discrimination, and these are shadowed by low rates of complaints. This renders the experiences of most people invisible to institutions and professionals. 

## 9. Neglect

The Royal Commission into Violence, Abuse, Neglect, and Exploitation of People with Disability defines “neglect” to include physical or emotional neglect, passive neglect, or wilful deprivation. Neglect can be a single significant incident or a systemic issue that involves depriving a person with disability of the basic necessities of life, such as food, drink, shelter, access, mobility, clothing, education, medical care, and treatment [48].

Failure to provide personal care and failure to communicate are acts of neglect. People who live with sensory loss often depend on others to provide personal care, information, and safe mobilisation. These are important acts that contribute to increasing the sense of safety, comfort, and well-being as well as empowering people with sensory loss with a degree of personal agency during their stay in the hospital. Most of the participants experienced neglect of their care and communication needs. However, the constrained question set did not allow for the details of what exactly was neglected to be ascertained, for example, interpreters providing communication support, nurses and aides helping with the activities of daily living, being alerted to food and drink arrivals, and orientation to bathroom layouts and locations. As the comorbidity of multiple disabilities is common in this population, care needs may be complex, and human assistance is critical. The Nordic definition explanatory notes alluded to earlier detail the many complexities of DBDSI and how it is the responsibility of society (and health institutions) to provide funding, care, and support. Professionals and hospitals must do more to avoid neglecting situationally vulnerable patients with DBDSI when they are in hospital.

## 10. Discrimination

If people who live with disability have the right to the highest attainable standard of health without discrimination according to international, federal, state, and territory laws, then this report card makes it evident that they are not receiving such care in practice. This study shows that patients with DBDSI are treated sub-optimally in Australian hospitals. They are unseen, unheard, and uncared for, and, therefore, they are also unsafe. More than three-quarters of the participants had no confidence that they were treated equally.

These participants with DBDSI do not need to be treated “equally” with non-DBDSI patients; they need to be treated equitably, with accommodations, extra care, and communication support, to achieve better health outcomes and ensure personal safety.

## 11. Administrative Interactions

One cannot be included or feel included if one is not welcome. The results show that 94% of participants felt unwelcome in their hospitals. A lack of seeking feedback and a complete failure to manage wait times further underline the exclusion by hospitals of participants with DBDSI. 

## 12. Low Rates of Complaint

All participants in all hospital types experienced sub-standard care and/or communication. A low rate of complaints (6%) implies that significant barriers exist to making complaints and giving feedback. Another explanation could be that there is a tacit acceptance of poor treatment. 

Research tells us that negative healthcare and hospital experiences are linked to poorer outcomes in health and well-being for all patients. For patients living with DBDSI, the risk of poorer outcomes is much greater with communication and care failures [42,49]. The paucity of literature on hospital experiences by this group shows that there are epistemic injustices, where these experiences are not solicited or made legible. This confers data invisibility. Patient experience measures and patient feedback methods need to be accessible with a human point of contact who can take responsibility to ensure people with DBDSI are given accessible-to-them ways to express their opinions and experiences. This will take time and require funding. In fact, the same provisions are needed for in-hospital communication encounters, true informed consent, querying care, shared decision-making, and making complaints. The same considerations apply for research projects in any domain if they are to be participatory and inclusive. These are, in fact, basic rights enshrined by existing protections and policies in Australia and in the UNCRPD [3]. 

## 13. Waiting, Waiting, Waiting

With respect to the management of waits and appointments, no participant felt their waits were well managed. That, in combination with poor communication and reduced access to information, suggests not knowing what is going on as the de facto “state of being” while in the hospital, and I wonder if this may have made waits seem longer. The qualitative inquiry component of the research will be needed to unpack this and will be presented in a separate article.

## 14. Consent Interactions

There is a universal failure to provide accessible consent forms. Coupled with the previously mentioned communication failures, this means that patients with DBDSI in any hospital type are not receiving the legislated requirement that all consent is truly informed. If patients with DBDSI are to have agency, they need to be informed about the benefits and risks of what is happening, what will happen, and how that will occur—information that enables genuinely informed consent.

People with DBDSI require accessible versions of this information to enable their decision-making on whether they choose to give consent to the medical procedure/treatment or to the proposed research. From the results of this research, it seems reasonable to infer that informed consent was never truly obtained from any participant. 

## 15. Where to Next?

The participants reported experiences from multiple different hospitals, so these findings suggest a broad culture of failing to provide patient-centred care and accessible-to-the-patient communication. Overall, the report card shows poor communication by professionals, dehumanisation of participants with DBDSI, evident discrimination, neglect of care needs, and the outright failure to provide a consent form accessible to any participant. These are not only legal failures but also show moral and ethical noncompliance with the standards expected by the community at large. The findings showcase the urgency for more research and remedial actions to be undertaken by both professionals and institutions.

It is clear from the data, however, that health professionals need more awareness of their legal responsibility to communicate with their patients in accessible-to-them ways. Obtaining informed consent requires sufficient communicative exchange so that the patient understands what is happening, why, and what the potential risks are. People with DBDSI sometimes have procedures they know nothing about. We need to translate this research into action that involves healthcare interaction for DBDSI. We could start using existing resources to provide accessible consent forms. A small step that would begin the process towards informed consent with DBDSI is knowing what the procedure/treatment is planned and knowing what the risks and benefits are. 

Therefore, work needs to begin with student learning and extend to professional practice for better communication and care. We also need to harness the capabilities of people with DBDSI themselves to be better prepared for hospitalisations, but the principal responsibility for communication and care falls on the professionals and caregivers. Hersh (2013) makes the valuable observation that it is not the sensory impairment that “causes” the person with DBDSI to not know what is going on but rather the healthcare provider’s failure to communicate effectively [35]. More research is needed into how best to achieve these significant and systemic changes. 

There are limitations to this research. While quantitative data give us the breadth of what is happening (as in the report card of communication and care failures above), qualitative data are imperative to examine how deep this turbulent water is. What are the barriers and enablers? In this research, data were not collected on the length of stay and did not encompass First Nations people’s experiences—this needs community-owned research. 

This report card flags the existence of communication and care inequities in Australian hospitals but does not reveal their depth, specificities, and/or complexities. A constrained question set does not capture experiences that are different from those being asked about and anticipated. From the transcripts of qualitative inquiry narratives told by the same participants with DBDSI about their many hospital experiences over two years, harrowing stories emerge of pervasive, widespread failures of care and communication—details that the AHPEQS in no way anticipated or allowed for. These will be discussed, as mentioned, in a separate qualitative inquiry article.

Notwithstanding the limitations of quantitative methodology, as noted above, several key issues have been identified from the findings of this research. While qualitative inquiry methods will be necessary to explore the nature of interactions, they will also help identify solutions by asking expert knowers themselves what is needed. Such salutogenic and positive framings enable expert knowers to construct solutions. Fricker (2011) attests that most knowledge about a system will come from those most impacted by those systems. This knowledge includes solutions [28]. 

From this survey, however, first and foremost, there is an imperative for attitudinal and cultural change within health institutions and the professionals that work there. Funding for this systemic change, professional upskilling, and better individual patient support is crucial. Hospitals already have the ability to print alternative formats of patient support literature, menus, consent forms, and discharge instructions. Therefore, it should be made mandatory. Compliance needs to be audited. Accessibility must be built into every form, document, and leaflet from the beginning. This way, no patient or worker with a communication disability (or a person who forgot their reading glasses) is excluded. And while we are talking about accessible formats, let us not forget plain English versions as well. Many people with sensory loss or a communication disability likely have lower educational attainment and less English proficiency. For example, cultural and linguistic diversity and/or coexisting cognitive impairments are not uncommon.

Attention to communication practices and strategies to improve professional competencies is crucial. Healthcare professionals must learn and use basic communication and assistive measures to reverse the alarming data presented here. We must make room in the curricula of health-related courses to consider the additional or more complex needs of patients with disability in our hospitals and our communities. Good communication practice should be imbued not just as stand-alone content but in every discussion with patients, participants, clients, and customers. Having both a communication difficulty and disability is common.

All encounters should begin, however, by asking people with DBDSI—the expert knowers—what is needed, what works, and what does not work for them. 

## 16. Conclusions

This study demonstrates that emancipatory and socially just research is possible and productive with people with DBSI, notwithstanding the extra people, time, cost, and complexity. The study was able to capture data from an often marginalised and excluded group because of three factors: the presence of a lived-experience researcher, cultural immersion and coproduction, and (critically) a commitment to accessibility for all.

The participants in this study are unified in their voicing of the few positive experiences and too many barriers. Underpinning these failures is the fact that they exist because hospitals, institutions, and professionals have the power to assist or ignore their patients.

Hospitals should safeguard and heal patients. From the evidence of this study, however, it is obvious they do not. This report card, based on the responses of participants with DBDSI to an adjusted standard hospital patient experience questionnaire, is grim reading. Patients living with disability experience a double assault on their sense of safety when in the hospital: first from the disease, surgery, or injury, and second from the power disparities that underpin neglect, reduced information, and a lack of care and communication. There is power at play in every service disparity and every piece of information not provided. These are occurring despite legal and policy “safeguards” and “assurances”. This report card speaks volumes about institutional and professional failures of care and communication with patients with DBDSI in Australian hospitals. They can, and must, do better.

## Data Availability

The ethics approval (ETH17-1398) precludes the sharing of this data, therefore the available data is used within this article.

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
