# Peer review of "Australian Hospital Experiences of People Living with Deafblindness or Dual Sensory Impairment: The Report Card"

_healthcare, 2024, doi:10.3390/healthcare12080852_

Round 1

Reviewer 1 Report

Comments and Suggestions for Authors

The article that has been presented deals with a subject of enormous relevance and, as it indicates, little studied. The introduction is very clear and goes into the essential aspects of the problem in depth. In addition, a clear effort is made to make everything indicated in the paper understandable and clear. 

The methodology is comprehensive and quite well delimited. The reference to epistemic justice in relation to the methodology seems to me to be an outstanding and brilliant element. On the other hand, it is stated that life histories have been conducted. In this case I understand that we could be talking about a qualitative approach, isn't it, at least in part? However, structured questions were also asked, so it would seem that there would also be a quantitative approach. Would we then be dealing with a mixed methodology? Everything leads me to believe that this is the case, but it would be good if the author of this paper could clarify this further.

I was very surprised by the results section, which is sparse and limited to quantitative results. I do not quite understand the reason for this. In the subsequent discussion we see the different categories that have been extracted from the life stories. So why are these categories not shown in the results? I mean, for example, the category of communication or the category of protective conventions. I suggest that a modification be made in this sense and that within each category some literal example be given of what the informants stated. Otherwise, it would appear that the categories are not supported by the qualitative evidence.

On the other hand, I also do not understand why the discussion section does not contain the analysis of the previous categories and why a new section has been created to analyse each category. I think it would be more appropriate to include sections 6-12 in the discussion. 

In point 13, the researcher previews a future article of a qualitative nature. I look forward to reading it. However, it would be necessary for some of this information to be transferred to this work in order to complete it. Right now I (humbly) consider that there is not enough evidence to support the assertions made.

Reviewer 2 Report

Comments and Suggestions for Authors

Dear Author(s),

I appreciate your paper. To improve it please find below my comments:

1. It is not clear the framework you adopted. Is the framework of shifting from cure to care? of public service design? of disability care? of patient experience/engagement?

2. Is the objective of the survey  to demonstrate the weakness of nurses, medical doctors and physicians and other heath worker professionals? For example is the aim to demonstrate that they need high communications skills/attitudes?

3. Having defined the objective of the research, I suggest to re-write the abstract since it appears too generic.

4. The paper is a little bit longer. I suggest to simplify. For example I consider some references to the general problem of disability redundant. The figures of WHO are interesting but they are not relevant for the paper.

5. In the discussion of results it will be helpful for the reader to see a relevance hierarchy.

6. Even if the content of the survey is patients' experience, it will be helpful to have some data about the services provided. I mean how many patients are treated? Are there waiting lists? Effectiveness or efficiency indicators

7. Which are the implications of the survey? To change heath workers framing? To change the patients' approach? To reorganize care processes? To define a patient 's bill of rights?

8. In conclusion, I suggest to summarize the conclusive results in a table showing the relations among different variables. I mean professional information, patients' expectations, patients' experience.

Reviewer 3 Report

Comments and Suggestions for Authors

Australian hospital experiences of people living with deaf- 2 blindness or dual sensory impairment: The report card

By Watharow

This study used mixed methods pre- COVID-19 pandemic in order to explore the experiences of those with deaf-blindness (DB) and dual sensory impairment (DSI) in Australian hospitals.   This study is important and reads well; however, I have major concerns about this paper:

1.    Abstract – DB and DSI are used but then you revert back to the full form in line 15, please be consistent. The same is true in line 175. 

2.    Abstract – there seems to be an extra space between words in line 19. Same issue in line 41, 177, 183, etc.  

3.    Introduction - Line 25 mentions “We” but the author seems to be a single individual, see also lines 45-50. In other places, “I” is used e.g. line 214.

4.    Introduction – Line 26 refers to sensory loss being very common but for who? Surely this refers to a particular vulnerable group? E.g. older adults or others? This is alluded to in Line 95, perhaps briefly mention it.  

5.    Introduction – - Line 35 has capitalized D in Deaf but other places lowercase d is used e.g. deafblindness; please keep consistency.

6.    Methodology – Where are the research questions that were developed for this study? What are the study aims?

7.    Methodology – Line 184 – minor issue but commas are missing in line 184.

8.    Methodology – Line 328 – can you tell us a little more about Usher syndrome?

9.    Methodology – Were financial or other incentives available to participants for participating in the research?

10.  Methodology – What were the theoretical frameworks that informed this study ?

11.  Methodology - How were the participants recruited?

12.  Methodology - How were they identified and why?

13.  Methodology - What is the qualitative method used?

14.  Methodology - How was the analysis done?

15.   Results - What is novel here?

16.   Results- Are there differences between groups?

17.   Discussion - What future research is needed?

18.  Discussion – Is the study generalizable?

Thank you for the opportunity to review this manuscript.

Comments on the Quality of English Language

Introduction - Line 25 mentions “We” but the author seems to be a single individual, see also lines 45-50. In other places, “I” is used e.g. line 214.

Introduction – - Line 35 has capitalized D in Deaf but other places lowercase d is used e.g. deafblindness; please keep consistency.

   Methodology – Line 184 – minor issue but commas are missing in line 184.

Reviewer 4 Report

Comments and Suggestions for Authors

This manuscript is an excellent and important contribution to the literature on disability and healthcare access. My comments are minor within the context of the work presented.

To start, this paper does a good job at describing the difficulty with language/terminology around deafblindness-dual sensory impairment. Given how this also may affect the ability of researchers to find relevant literature, this issue has implication for moving research forward on this topic so that researchers can even locate work on this topic. The author is well-situated as both a researcher and PWDBDSI to put forth a term. Two small points (1) the author uses the term “we” repeatedly throughout the discussion of terminology on page 2. As there is just a single author, this should be updated using an approach consistent with this journal’s editorial guidelines; (2) the acronym itself if quite long and the author misuses it. See PWDBDDSI on line 406, PLWDBSI on line 564 and PWDBSI on line 600. I do not wish to recommend the author consider a shorter acronym, but I point this out to rectify in text and to consider how it may result in typos by future authors.

The context provided in the background section on the size of the population and the related issue of the aging population was useful. The methodological section in particular does an excellent job of describing the previous omissions of PWDBDSI and the specific challenges of collecting data from this population. The outline of the methodological difficulties experienced and accommodations made were well described (and justified).

One thing missing from the methods section is a description of how the 18 participants were recruited into the study. What technique was used to sample?  Also, what is known of these individuals’ hospital visits in total? Did all of them visit a hospital at least once during the reference period?  More detail on this would be useful to provide context to the study findings. I think this is particularly an issue as it relates to the title, referring to this as a “report card.”  What hospitals are represented? If this is meant to be a report card for hospitals broadly, in how they provide care for PWDBDSI, I think that point can be made (and is made). However, it would be helpful to spell this out a bit in detail (e.g., “The participants received care at X different Australian hospitals, suggesting that this is a broad issue”?)

On the point of the findings, these are well discussed, with useful recommendations that follow from such stark outcomes. I find this to be well-written and a necessary contribution to this literature.

Round 2

Reviewer 1 Report

Comments and Suggestions for Authors

The changes made have improved the work. Now I think it has more consistency and more depth. The introduction is more appropriate. The methodology is much clearer. Congratulations on the final work and thank you for the effort made.

Reviewer 3 Report

Comments and Suggestions for Authors

This study used a larger mixed methods pre- COVID-19 pandemic in order to explore the experiences of those with deaf-blindness (DB) and dual sensory impairment (DSI) in Australian hospitals.   This study is important and reads well; and the author has responded to the queries and made the relevant revisions. It can be accepted for publication. 

Thank you for the opportunity to review this research.